# Comparison of Environmental Conditions on Summits of Mount Everest and K2 in Climbing and Midwinter Seasons

**DOI:** 10.3390/ijerph18063040

**Published:** 2021-03-16

**Authors:** Robert K. Szymczak, Michał K. Pyka, Tomasz Grzywacz, Michał Marosz, Marta Naczyk, Magdalena Sawicka

**Affiliations:** 1Department of Emergency Medicine, Faculty of Health Sciences, Medical University of Gdansk, Debinki 1, 80-211 Gdansk, Poland; 2Polish Mountaineering Association, Polish National K2 Winter Expedition 2018 Support Team, Mokotowska 24, 00-561 Warszawa, Poland; marta.naczyk@gumed.edu.pl; 3Institute of Physical Culture, Kazimierz Wielki University in Bydgoszcz, Chodkiewicza 30, 85-064 Bydgoszcz, Poland; tomgrzyw@ukw.edu.pl; 4Institute of Meteorology and Water Management—National Research Institute, Waszyngtona 42, 81-342 Gdynia, Poland; michal.marosz@imgw.pl; 5Department of Nutritional Biochemistry, Faculty of Health Sciences, Medical University of Gdansk, Debinki 1, 80-211 Gdansk, Poland; 6Department of Neurology, Faculty of Medicine, Medical University of Gdansk, Debinki 1, 80-211 Gdansk, Poland; magdalena.sawicka@gumed.edu.pl

**Keywords:** winter sports, mountaineering, environmental conditioning, extreme altitude, Mount Everest, K2

## Abstract

(1) Background: Today’s elite alpinists target K2 and Everest in midwinter. This study aimed to asses and compare weather at the summits of both peaks in the climbing season (Everest, May; K2, July) and the midwinter season (January and February). (2) Methods: We assessed environmental conditions using the ERA5 dataset (1979–2019). Analyses examined barometric pressure (BP), temperature (Temp), wind speed (Wind), perceived altitude (Alt), maximal oxygen uptake (VO_2_max), vertical climbing speed (Speed), wind chill equivalent temperature (WCT), and facial frostbite time (FFT). (3) Results: Most climbing-season parameters were found to be more severe (*p* < 0.05) on Everest than on K2: BP (333 ± 1 vs. 347 ± 1 hPa), Alt (8925 ± 20 vs. 8640 ± 20 m), VO_2_max (16.2 ± 0.1 vs. 17.8 ± 0.1 ml·kg^−1^·min^−1^), Speed (190 ± 2 vs. 223 ± 2 m·h^−1^), Temp (−26 ± 1 vs. −21 ± 1°C), WCT (−45 ± 2 vs. −37 ± 2 °C), and FFT (6 ± 1 vs. 11 ± 2 min). Wind was found to be similar (16 ± 3 vs. 15 ± 3 m·s^−1^). Most midwinter parameters were found to be worse (*p* < 0.05) on Everest vs. K2: BP (324 ± 2 vs. 326 ± 2 hPa), Alt (9134 ± 40 vs. 9095 ± 48 m), VO_2_max (15.1 ± 0.2 vs. 15.3 ± 0.3 ml·kg^−1^·min^−1^), Speed (165 ± 5 vs. 170 ± 6 m·h^−1^), Wind (41 ± 6 vs. 27 ± 4 m·s^−1^), and FFT (<1 min vs. 1 min). Everest’s Temp of −36 ± 2 °C and WCT −66 ± 3 °C were found to be less extreme than K2’s Temp of −45 ± 1 °C and WCT −76 ± 2 °C. (4) Conclusions: Everest presents more extreme conditions in the climbing and midwinter seasons than K2. K2’s 8° higher latitude makes its midwinter BP similar and Temp lower than Everest’s. K2’s midwinter conditions are more severe than Everest’s in the climbing season.

## 1. Introduction

The history of alpinism shows that the altitudes considered by one generation to be at the limit of human survival have been exceeded by the best climbers of succeeding generations [1]. In May 1978, R. Messner and P. Habeler achieved the final altitudinal limit of 8848 m Mount Everest (Everest) without supplemental oxygen [2,3]. Their ascent proved the thesis propounded in 1920 by Alexander Kellas, the pioneer of high-altitude physiology, that Everest could be ascended by humans of extreme physical and mental constitution without supplementary oxygen [4]. In September 1978, L. Reichardt climbed K2 (8611 m), the second highest 8000 m (8K) peak, without oxygen.

Reaching the top of the world without supplemental oxygen has encouraged mountaineers to search for more extreme high-altitude challenges. Since Messner and Habeler’s success, Everest and K2 had each been climbed more than 200 times without oxygen by 2021 [5,6,7]. Winter Himalayan mountaineering was launched in February 1980 with the first winter oxygen-supported ascent of Everest by K. Wielicki and L. Cichy [8]. Only one climber, Ang Rita Sherpa, has repeated their feat without supplemental oxygen—on the first day of winter, 22 December 1987 [5,9,10]. By 2021, all 8K peaks but K2 had been climbed in winter without oxygen [5]. K2 was finally conquered by a team of 10 Nepalese climbers on 16 January 2021 [11]. One climber, Nirmal Purja, reached the summit without supplemental oxygen [12].

Apart from the elite mountaineers who focus on climbing Everest and K2 in midwinter without supplemental oxygen, a growing number of commercial expeditions are interested in high-altitude winter expeditions [13].

Meteorological factors that limit human survival at the top of Everest—such as barometric pressure (BP), wind chill equivalent temperature (WCT), and facial frostbite time (FFT)—have already been assessed with meteorological datasets and in situ observations [9,14,15,16,17,18,19,20,21,22,23,24]. The estimated average monthly BP in May on Everest, its main climbing season, is 333 hPa, and in midwinter in January and February, it is 323 hPa [23]. The average climbing-season WCT is −45 °C, and the average climbing-season FFT is 7 min; the average midwinter WCT is close to −65 °C, and the average midwinter FFT is less than one minute [20]. The most extreme air temperature (Temp) on the summit of Everest might be as low as −49 °C, and wind speed (Wind) might be as high as 80 m·s^−1^ [24].

Surprisingly, similar calculations had not yet been made for K2 (8611 m), which is the highest peak in the Karakoram range. K2 is the second highest peak in world and was the last 8K peak climbed in winter. Latitude and altitude are the main factors that determine climatic parameters such as Temp, BP, and Wind [25]. The higher the altitude and the latitude and the colder the season, the lower are BP and Temp at a given altitude [14,25,26,27,28]. The summit of K2 is 237 m lower than Everest’s, but it is located 8° further north, which might substantially affect its weather conditions, especially in midwinter. Therefore, this study aimed to assess and compare weather at the summits of Everest and K2 in the main climbing season and in midwinter, as well as to precisely define which summit has the most extreme environmental conditions and in which season that occurs.

We analyzed the climatic parameters most limiting human performance and survival at high altitudes. BP determines the partial pressure of inspired oxygen, which is critical for physiological performance, maximum oxygen uptake (VO_2_max), and the speed of vertical ascent (Speed) at extreme altitude [1,23,29,30,31]. A combination of low Temp and high Wind in parameters such as WCT and FFT determines the risk of hypothermia and frostbite [32].

## 2. Materials and Methods

We estimated environmental conditions at the summits of Everest and K2 in all months from 1979 to 2019, and then we conducted a comparative analysis of the climbing season (Everest, May [5,23]; K2, July [7]) and the midwinter season (January and February). The main climbing season on Everest is mostly restricted to May [5,23] because of summer snowfalls caused by monsoons [17,18,20]. K2’s climbing season is in July [7].

There are no weather stations on K2 that could provide in situ data. Therefore, our primary source of meteorological conditions was the ERA5 Reanalysis, a state-of-the-art database provided by the European Centre for Medium-Range Weather Forecasts (ECMWF) [33]. ERA5 is based on a model that assimilates all available observations, including satellite data, and is available at the high spatial resolution of 0.25 × 0.25 degrees of latitude and longitude. The calculated values of weather conditions should be interpreted as the average for a roughly rectangular area with sides of about 28 by 24 km (Everest) or by 22 km (K2) at the altitude of the summit in which the mountain is located.

In this research, four meteorological variables were used at selected isobaric levels: 300, 350, and 400 hPa. We used hourly means of geopotential height, air temperature, and wind vector (u, zonal; v, meridional) to calculate Wind. The data preparation required us to calculate BP at the summit levels of Everest and K2, with a subsequent interpolation of Temp and Wind. We calculated BP by fitting a nonlinear regression model to each instance of the air pressure–height profile. Calibrated model coefficients were then used to estimate BP at a given height, thus providing the data we needed to estimate the value of other variables. Temp and Wind were derived from linear interpolation using values at predefined isobaric surfaces from ERA5 and the calculated value of BP at summit level. The temperature lapse rate changes with the season [34], so Temp was calculated individually for each instance. The unit of analysis we used for each weather parameter was the monthly mean based on a sample of 41 monthly means between 1979 and 2019.

BP on the summits of Everest and K2 was presented as perceived altitude (Alt), which was calculated using the model atmosphere equation [27]:BP [hPa] = ln(6.63268 − 0.1112 h − 0.00149 h^2^) 0.75^−1^, (1)
where h is a perceived altitude in [km] [27]. This equation is representative for the pressure–altitude relationship at a location within 30° of the equator in summer months, so perceived altitude is the altitude at which the given BP would exist in summer months within 30° of the equator.

We also presented BP values in terms of VO_2_max and Speed using the formulas presented by Matthews et al. [23], who synthesized the conclusions of other authors [30,31,35,36]:VO_2_max [mL·kg^−1^·min^−1^] = [ln(PiO_2_ 0.750) − 3.25] 0.0308^−1^,(2)
where PiO_2_ is a partial pressure of inspired oxygen in hPa at a given altitude; and
Speed [m·h^−1^] = [41.54 85% VO_2_max − 255.96] 60.(3)

The results of VO_2_max and Speed are representative for a male Caucasian climber with a sea-level VO_2_max of about 57 mL·kg^−1^·min^−1^ [14,35]. For the calculation of Speed, we set the mass of a hypothetical climber with equipment at 100 kg [14,30].

From the Temp and Wind results, we estimated the thermal stress indices WCT and FFT. Our WCT and FFT calculations we based on formulas by Osczevski and Bluestein and by Tikuisis and Osczevski [37,38]:WCT [°C] = 13.12 + 0.621T − 11.37V^0.16^ + 0.3965TV^0.16^(4)
FFT [min] = [−24.5(0.667V + 4.8) + 2111] (−4.8 − T)^−1.668^,(5)
where V is the wind speed at 10 m above the surface (km·h^−1^) and T is the air temperature (°C).

### Statistical Analysis

We used Statistica 13.1 (StatSoft, Tulsa, OK, USA) for our calculations, testing the normality of data with the Shapiro–Wilk W-test and then the homogeneity of variance with the Brown–Forsythe test. To assess the statistical significance of differences, we used analysis of variance and Tukey’s post-hoc test. Our results are expressed as means with a standard deviation (x ± SD). We measured the relative variability of all parameters using a coefficient of variation (CV) and analyzed any associations of the parameters using Pearson’s correlation coefficient (r). We set statistical significance at *p* < 0.05 for all our analyses.

## 3. Results

### 3.1. Barometric Pressure

The average climbing-season BP on Everest’s summit was found to be 13 ± 1 hPa lower than on K2 (*p* < 0.001). The BP calculated for the climbing season corresponded with a perceived altitude (Alt) of 8925 m on Everest and 8640 m on K2 (285 m difference); see Table 1 and Figure 1.

Midwinter BP values on both peaks were lower than in the climbing season (*p* < 0.001). Everest’s midwinter average BP was 2 ± 1 hPa lower than on K2 (*p* < 0.001). The climbing season BP difference of 13 ± 1 hPa between both peaks was much higher than in midwinter. The drop of BP between the climbing and the midwinter season was higher on K2 (21 ± 2 hPa) than on Everest (9 ± 1 hPa). Winter BP corresponded with a perceived altitude of 9134 ± 40 m on Everest and 9095 ± 48 m on K2 (only 39 m difference). The midwinter VO_2_max difference between both peaks was only 0.2 ± 0.2 mL·kg^−1^·min^−1^ (*p* < 0.001), much less than 1.6 mL·kg^−1^·min^−1^ (*p* < 0.001) in the climbing season. The midwinter Speed difference between both peaks was much lower than during the climbing season, 5 ± 3 (*p* < 0.001) vs. 33 ± 3 m·h^−1^ (*p* < 0.001); see Table 1 and Figure 1.

K2’s midwinter BP was 8 ± 2 hPa lower than Everest’s in the climbing season (*p* < 0.001). The perceived altitude of K2’s summit in midwinter was 170 m higher than Everest’s in May (9095 ± 48 vs. 8925 ± 20 m). VO_2_max and Speed were significantly lower in midwinter on K2 than in the climbing season on the summit of Everest; see Table 1 and Figure 1.

### 3.2. Air Temperature

K2’s summit Temp was significantly lower than Everest’s in all the corresponding months of the year. However, the average climbing season Temp in May on Everest’s summit was 6 ± 2 °C lower than in July on K2 (*p* < 0.001). In midwinter, Temp was 9 ± 2 °C lower on K2 than on Everest (*p* < 0.001). The reduction of Temp between the climbing and midwinter seasons was higher on K2 (25 ± 2 °C) than on Everest (10 ± 2 °C); see Table 2 and Figure 2.

### 3.3. Wind Speed

The average climbing season Wind on the Everest’s summit was only 1.1 ± 4.1 m·s^−1^ higher than on K2 (*p* = 0.99996). Higher midwinter Wind was encountered on both peaks: the average on Everest was 14 ± 5 m·s^−1^ higher than on K2 (*p* < 0.001). The increase of Wind between the climbing and midwinter seasons was higher on Everest (25 ± 6 m·s^−1^) than on K2 (12 ± 4 m·s^−1^); see Table 2 and Figure 3.

### 3.4. Wind Chill Temperature and Facial Frostbite Time

The average climbing-season WCT on Everest’s summit was 8 ± 3 °C lower than on K2 (*p* < 0.001). In contrast, the average midwinter WCT on K2 was 10 ± 2 °C lower than on Everest (*p* < 0.001). The drop in WCT between the climbing and midwinter seasons was higher on K2 (39 ± 3 °C) than on Everest (21 ± 2 °C).

The average climbing-season FFT on Everest was 5 ± 3 min shorter than on K2 (*p* < 0.001). FFT in winter was extremely low on both peaks but shorter on Everest (<1 min) than on K2 (1 min) (*p* < 0.001); see Table 2.

### 3.5. Combined Weather Parameters

We found a strong positive correlation of BP and Temp (r ≥ 0.8; *p* > 0.05), a fairly negative correlation of BP and Wind (Everest r = −0.3; K2 r = −0.5; *p* < 0.05), and no correlation between Temp and Wind on both peaks in the climbing season.

In midwinter, BP positively correlated with Temp: on K2, the correlation was strong (r ≥ 0.8; *p* > 0.05), but on Everest, it was fair to moderate (February r = 0.4; January r = 0.6; *p* > 0.05). There was no correlation between BP and Wind in the midwinter months on both peaks. In midwinter, Wind moderately positively correlated with Temp on Everest (r ≥ 0.6; *p* < 0.05) and fairly positively correlated with Temp in January on K2 (r = 0.3; *p* < 0.05).

## 4. Discussion

### 4.1. Barometric Pressure

Our research aimed to compare climbing conditions on the two summits, but no in situ data were available for K2, so we presented calculations derived from ERA5. Everest summit’s estimated mean monthly BP of 333 ± 1 hPa in May, its climbing season, and 324 ± 2 hPa in midwinter corresponded with calculations by other authors: 335 and 324 hPa calculated from radiosonde data [14] and 333 and 323 hPa estimated from in situ measurements combined with data from the ERA5 Reanalysis [23]. Matthews et al. corrected data derived from ERA5 with in situ measurements from automatic weather stations installed on Everest’s South Col (7945 m) and Balcony (8430 m) [23]. The differences between these results and ours were minimal: Everest’s May BP was the same, and the midwinter value was only 1 hPa higher in our study compared with Matthews et al. [23].

BP decreases with higher altitude and latitude, but latitude has a much greater effect in midwinter [25,27]. The relatively large difference between BP on both peaks in climbing season was caused not only by the altitudinal difference between both peaks but also by the different months for the main climbing season on both mountains. K2’s season is in July, the month when a BP of 347 ± 1 hPa was the highest of the year. On Everest, the mean BP of 338 ± 0.5 hPa in July was also one of the highest of the year. However, summer snowfalls on Everest caused by monsoons restrict the climbing season mainly to May, when BP was found to be 333 ± 1 hPa, which was 5 ± 1 hPa lower than in July. Apart from the 237 m of altitude difference between peaks, the less barometrically favorable climbing month for Everest was responsible for the huge BP discrepancy between the peaks in climbing season.

The difference between midsummer and midwinter BP at a given altitude increases with higher latitude [27], which is in line with our results of a significantly larger BP difference between midwinter January and midsummer July on K2 (21 ± 2 hPa: 325 ± 2 vs. 347 ± 1) than on Everest (15 ± 12 hPa: 324 ± 1 vs. 338 ± 0.5). K2’s latitude (35°52′ N) is 8° further north than Everest’s (27°59′ N). K2’s latitude has another important implication: latitudes between 30° and 45° N, where K2 is located, show the biggest difference between the midwinter and midsummer latitudinal effect on BP. Between latitudes of 30° and 45° N, at a given altitude BP drops three times more in midwinter than in midsummer [27]. In the midwinter season, the 8° higher latitude of K2 largely reduced the effect of its 237 m lower altitude on BP, making BP higher but very close to that of Everest in midwinter (2 ± 1 hPa difference: 326 ± 2 vs. 324 ± 2) and significantly lower than on Everest in climbing season (8 ± 2 hPa difference: 326 ± 2 vs. 333 ± 1 hPa).

The midwinter BP values we estimated for Everest (324 ± 2 hPa) and K2 (326 ± 2 hPa) were higher than the BP values of 315 hPa [15] and 302 hPa [23,31] identified as the probable performance borderlines where VO_2_max is significantly reduced. Some authors hold that Bailey’s estimate of 302 hPa was incorrectly based on a linear regression model and is unrealistic due to the significantly reduced climbing rates at extreme altitude [39]. Still, the lower BP in midwinter on both peaks observed in our study was physiologically and practically significant because of its effect on VO_2_max and Speed. The 9 ± 1 hPa BP difference between climbing season and midwinter on Everest amounted to a 7% reduction in VO_2_max and a 13% decrease in Speed. The 21 ± 1 hPa BP difference on K2 amounted to a 14% reduction in VO_2_max and a 24% decrease in Speed in a midwinter summit bid. The longer ascent and descent times caused by the slower climbing pace in midwinter should be considered in summit bid logistics, such as setting the altitude of the last camp and the times of the start of and withdrawal from a summit bid. An additional unfavorable circumstance in planning a midwinter summit attack is the shorter winter days.

The prediction of the future BP levels on the summits of Everest and K2 should consider they will likely increase due to global climate warming. HadCRUT4, a new dataset of global and regional temperature evolution, demonstrates that global mean annual ambient temperature increased at a rate of 0.17 °C a decade from 1979 to 2010 [40]. Global warming has increased the annual mean BP near Everest’s summit at a rate of 0.35 hPa a decade based on ERA5 Reanalysis from 1979 to 2019 [23] and 0.2–0.3 hPa a decade based on National Centers for Environmental Prediction (NCEP) Reanalysis from 1948 to 2008 [41].

The perceived altitude of both peaks in climbing season is higher than their altitude above sea level (a.s.l.). The model atmosphere equation for the pressure–altitude relationship that we used in our research was originally calculated to predict BP at most sites of interest to high-altitude medicine and works well for locations within 30° of the equator in midsummer [27,28]. The main climbing month on Everest is May, when BP is lower than in mid-summer months. The Alt in May (8925 ± 20 m) is higher than Everest’s altitude a.s.l. (8848 m). The perceived altitude on the summit of Everest in June equals its altitude a.s.l, which proves the model’s atmosphere formula to be correct. K2’s latitude (35°52′ N) is higher than 30°; therefore, because BP at a given altitude decreases with latitude, BP on K2’s summit in July is lower than if it were located at a lower latitude within 30° of the equator. This explains how K2’s Alt in July was found to be 30 m higher than its altitude a.s.l. (July 8640 ± 20 m). When extremely low midwinter summit BP values are expressed as perceived altitude, Everest and K2 are perceived as 9000 peaks in winter, thus making them the only two such extreme altitudes on Earth.

The same model atmosphere equation was used by Huey and Ward to determine present-day PiO2 at 9000 m [39]. The estimated value was used as a minimum level of tolerable PiO2 for calculations of maximum altitude reachable by mountaineers over geological time. Matthews et al. also converted the differences in Everest’s BP to changes in elevation [23]. However, they assumed that in May at a pressure of 333 hPa, the perceived altitude was equal to Everest’s height a.s.l. (8848 m), not the 8933 m from the model atmosphere equation. This explains why their corresponding Alt values were lower than ours by about 85 m.

### 4.2. Air Temperature

Our results showed that K2’s 8° further north than Everest’s latitude affects Temp more than its 237 m lower altitude and makes its Temp significantly lower than Everest’s throughout the year, especially in midwinter. The main reason that Everest’s Temp is lower in climbing season is that the climbing season is in spring (May) rather than in summer, thus accounting for the lower temperatures (Figure 2).

### 4.3. Wind Speed

Though Wind is similar on K2 and Everest in the climbing season, in midwinter, Everest experiences significantly higher Wind than K2. This corresponds with global jet stream trends analyzed by other authors [42,43]. The Northern Hemisphere jet stream forms a band between northern Africa and Hawaii in winter, but it weakens and shifts northward in summer [42]. This explains why Wind on K2 and Everest in the climbing season was found to be significantly lower than in winter. In winter, the Northern Hemisphere jet flow splits over the Atlantic Ocean into the Subtropical jet stream (STJ) flowing from western Africa to the western Pacific and the Polar Front jet stream (PFJ) flowing over northern Europe and Eurasia [43]. At the longitude of the Himalayas and Karakoram, the STJ flows between 20° N and 35° N, above the Himalayas; the PFJ flows between 40° N and 70° N [43]. This leaves a 5° gap between 35° N and 40° N that lies over Karakoram, which would explain the significantly lower winter Wind on K2 (Karakoram) that we calculated compared with Everest (Himalayas).

The STJ shifts towards the pole by about 0.1° a decade [43]. K2 is on the northern edge of the STJ, so this northward shift will make future K2 winter ascents more difficult.

The wind speed is a weather parameter that is commonly used to identify a suitable weather window for an ascent. Everest’s mean monthly midwinter Wind is about 50% higher than on K2, so the prospect of a proper acclimatization schedule and finding an acceptably low Wind weather window in midwinter are definitely diminished on Everest.

### 4.4. Wind Chill Temperature and Facial Frostbite Time

Our results showed that climbing season WCT and FFT values are lower on Everest, but in midwinter, WCT is lower on K2 and FFT is shorter on Everest. The colder and windier main climbing season on Everest explained its lower values of WCT and FFT than those for K2. The lower midwinter WCT on K2 than on Everest was explained by the lower Temp on K2. WCT depends more on ambient temperature than on wind speed [37], so the higher midwinter Wind that we calculated on Everest had less effect on WCT values than the lower K2’s summit Temp. The WCT index is limited to wind speeds below 27 m·s^−1^ [37], so the effect of extremely high midwinter Wind on WCT at K2, and especially at Everest, could be underestimated. The skin’s rate of cooling is more sensitive to wind speed than to air temperature [38], so higher midwinter Wind on Everest explained its lower FFT than that on K2. However, FFTs below 0 min signal that the extreme combined Temp and Wind midwinter conditions on Everest and K2 might be beyond the scope of the FFT index.

Everest’s WCT and FFT for the climbing and midwinter seasons that we estimated were similar to the WCT of −45 °C and FFT of 7 min in summer and WCT of −65 °C and FFT <1 min in winter assessed with the US NCEP Reanalysis data [20]. The NCEP Reanalysis is one of the earliest meteorological datasets. It has a 10-times-lower spatial resolution of 2.5 × 2.5 degrees of latitude and longitude compared with the ERA5 dataset, the highest quality reanalysis available at this time and what we used in our calculations [44].

Reduced air density at high altitude decreases convective heat loss. Estimates published by Huey et al. calculated that with the 60% decline of air density from sea level to 9000 m, the convective heat loss at 9000 m decreased by about 45% compared with those at sea level in the same conditions at a Temp of −33.5 °C and wind speeds up to 28 m·s^−1^ [45]. The standard equations for WCT and FFT [37,38] applied in our study and others [20] include the sea-level densities of air, which therefore might overestimate heat loss at altitude. Considering the estimates of Huey et al. [45], our results for WCT and FFT should be reduced to show more realistic values of heat loss at high altitude. The validation of WCT and FFT or a more complete human heat balance index, such as the Universal Thermal Climate Index (UTCI) [46], for the high-altitude, low barometric pressure environment would benefit future high-altitude thermal stress research.

### 4.5. Combined Weather Parameters

The positive correlation of BP and Temp that we observed for most months corresponded with the interrelation of both parameters described by the ideal gas law relating pressure, volume, temperature, and the amount of substance of a gas [27]. In the midwinter and climbing seasons, a higher BP accompanied a higher Temp. The fairly negative correlation of BP and Wind in the climbing season benefits climbers because it means that a climber should expect a higher BP in a low Wind weather window. The positive correlation of BP with Temp and the negative correlation with Wind that we calculated for both peaks in the climbing season supported the proposal by Moore et al. for Everest climbers that summit BP can act as a simple and easily implementable predictor of hypothermia and frostbite [20,22].

One in 34 climbers using supplemental oxygen dies during descents from Everest compared with one in 12 without oxygen support on Everest and one in five on K2 [6]. Supplemental oxygen reduces the risk of death [6,32,45], high-altitude deterioration, and hypothermia [47] at extreme altitudes. In view of the more adverse environmental conditions in the main climbing season on Everest than on K2, the technically more difficult route to the summit of K2 likely accounts for its higher death toll.

In midwinter, Wind moderately correlated positively with Temp on Everest, which means that during most days of a low Wind weather window, Temp would also be lower. This link was less evident on K2, where only in January did Wind correlate fairly with Temp. The higher temperatures that correlated with higher Wind in midwinter on Everest and K2 were likely related to the high winter STJ activity over this region, as described by others [43]. The higher correlation of Wind and Temp in midwinter on Everest than on K2 can be linked to the higher winter STJ activity over the Himalayas than over Karakoram.

The combined effects of extremely low BP (which significantly decreases VO_2_max and climbing pace), low Temp, and high Wind in midwinter will put climbers trying to reach the summit of Everest or K2 at high risk of frostbite, hypothermia, metabolic deterioration, and death [6,32,45,47,48,49].

We found that midwinter environmental conditions were more severe on Everest, mostly because of higher Wind, which would make acclimatization and a summit bid almost impossible. However, when weather risks were compared in potential low Wind weather windows, they appeared to be slightly worse on K2 because of its lower winter Temp and extremely low BP compared with Everest. Additionally, K2’s higher death statistics in climbing season, its more significant decline of weather conditions from the climbing season to midwinter, and its more technically difficult standard route suggested a higher risk of deterioration and death in midwinter summit attempts on this peak.

### 4.6. Limitations and Strengths

Our calculations of weather parameters were not based on in situ measurements but on the ERA5 Reanalysis dataset, which has a spatial resolution of 0.25 degrees of latitude and longitude.

This study provides the first comparison of weather parameters at the summits of K2 and Everest in the climbing season and in midwinter. These analyses might help future winter expeditions to maximize their chances of reaching the summits of K2 and Everest while minimizing the risks of their attempts.

## 5. Conclusions

We found that Everest’s higher altitude and spring climbing month presented more extreme climbing-season weather than the lower K2 with its climbing season in summer; K2’s 8° higher latitude was the main factor responsible for reducing its midwinter BP to values similar to Everest’s and to a significantly lower Temp than Everest’s; Everest’s extremely high midwinter Wind made its environmental conditions more severe and climbing more difficult than on K2 in midwinter. All K2’s winter conditions, including BP, were more extreme than Everest’s in climbing season.

## Figures and Tables

**Figure 1 ijerph-18-03040-f001:**
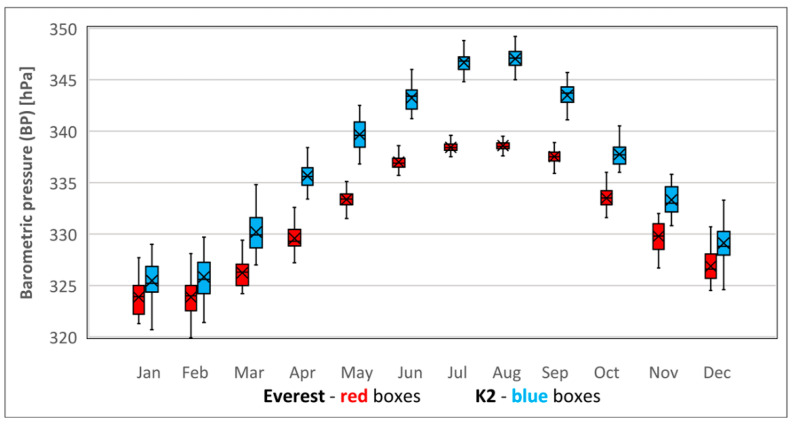
Annual cycle of barometric pressure (BP) at the summits of Everest (red boxes) and K2 (blue boxes). Sample size—41 monthly means (1979–2019). Bars extend to the maximum and minimum BP, boxes span the 25th–75th percentiles, crosses inside the boxes mark the monthly mean, and vertical lines inside the boxes represents the monthly median. BP values for all the corresponding months on the two summits were found to differ significantly (*p* < 0.001).

**Figure 2 ijerph-18-03040-f002:**
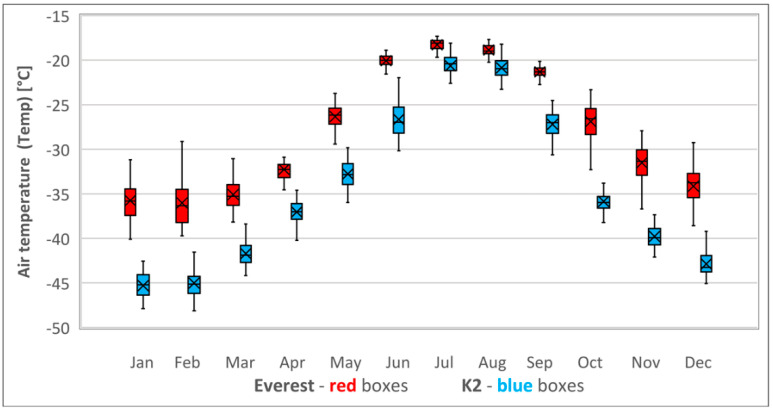
Annual cycle of air temperature (Temp) at the summits of Everest (red boxes) and K2 (blue boxes). Sample size—41 monthly means (1979–2019). Bars extend to the maximum and minimum BP, boxes span the 25th–75th percentiles, crosses inside the boxes mark the monthly mean, and vertical lines inside the boxes represent the monthly median.

**Figure 3 ijerph-18-03040-f003:**
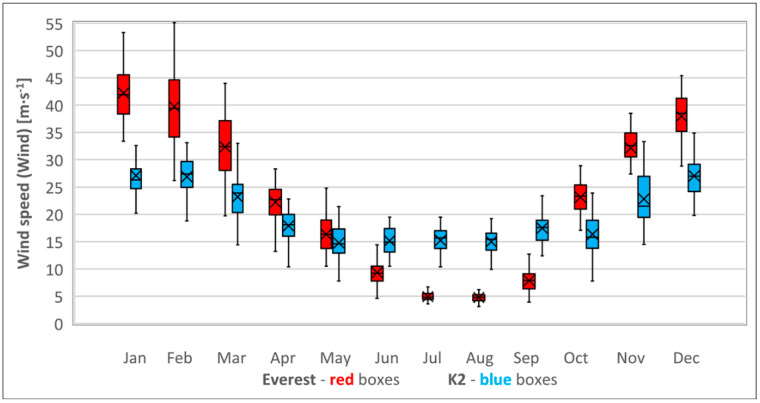
Annual cycle of wind speed (Wind) at the summits of Everest (red boxes) and K2 (blue boxes). Sample size—41 monthly means (1979–2019). Bars extend to the maximum and minimum BP, boxes span the 25th–75th percentiles, crosses inside the boxes mark the monthly mean, and vertical lines inside the boxes represent the monthly median.

**Table 1 ijerph-18-03040-t001:** Barometric pressure and BP-dependent parameters for the months we analyzed: monthly means ± standard deviation (SD), coefficient of variation (CV%), and minimum and maximum values. Sample size—41 monthly means (1979–2019).

	BP [hPa]	Alt [m]	VO_2_max [ml·kg^−1^·min^−1^]	Speed [m·h^−1^]
**Everest (May)**	333 ± 1 (0.3%)	8925 ± 20 (0.2%)	16.2 ± 0.1 (0.7%)	190 ± 2 (1%)
	min 331, max 336	min 8878, max 8972	min 16.0, max 16.5	min 184, max 195
**K2 (July)**	347 ± 1 (0.3%)	8640 ± 20 (0.2%)	17.8 ± 0.1 (0.6%)	223 ± 2 (1%)
	min 345, max 349	min 8595, max 8679	min 17.6, max 18,0	min 218, max 228
**Everest (January)**	324 ± 1 (0.5%)	9134 ± 39 (0.4%)	15.1 ± 0.2 (1.4%)	165 ± 5 (2.8%)
	min 321, max 328	min 9049, max 9192	min 14.7, max 15.5	min 159, max 175
**K2 (January)**	325 ± 2 (0.6%)	9099 ± 42 (0.5%)	15.3 ± 0.2 (1.5%)	170 ± 5 (2.9%)
	min 321, max 329	min 9020, max 9205	min 14.7, max 15.7	min 157, max 179
**Everest (February)**	324 ± 2 (0.6%)	9134 ± 41 (0.4%)	15.1 ± 0.2 (1.5%)	165 ± 5 (3.0%)
	min 320, max 328	min 9040, max 9223	min 14.6, max 15.6	min 155, max 176
**K2 (February)**	326 ± 2 (0.7%)	9091 ± 53 (0.6%)	15.3 ± 0.3 (1.9%)	170 ± 6 (3.7%)
	min 321, max 332	min 8952, max 9189	min 14.7, max 16.1	min159, max 187

BP, barometric pressure; Alt, perceived altitude; VO_2_max, maximal oxygen uptake; Speed, vertical climbing speed.

**Table 2 ijerph-18-03040-t002:** Ambient temperature, Wind, and thermal stress indices: WCT and FFT for the months we analyzed: monthly means ± standard deviation (SD), coefficient of variation (CV%), minimum and maximum values. Sample size—41 monthly means (1979–2019).

	Temp [°C]	Wind [m·s^−1^]	WCT [°C]	FFT [min]
**Everest (May)**	−26 ± 1 (5.1%)	16 ± 3 (19.4%)	−45 ± 2 (5.5%)	6 ± 1 (22.4%)
	min −29, max −24	min 11, max 25	min −50, max −41	min 3, max 9
**K2 (July)**	−21 ± 1 (6.6%)	15 ± 3 (17.3%)	−37 ± 2 (6.4%)	11 ± 2 (22.4%)
	min −24, max −18	min 8, max 20	min −42, max −31	min 7, max 18
**Everest (January)**	−36 ± 2 (6.1%)	42 ± 5 (12.1%)	−66 ± 3 (4.2%)	−2 ± 1 (68.0%)
	min −40, max −31	min 33, max 53	min −72, max −60	min −4, max 0
**K2 (January)**	−45 ± 1 (2.8%)	27 ± 4 (15.5%)	−76 ± 2 (2.6%)	1 ± 1 (63.3%)
	min −48, max −43	min 20, max 38	min −82, max −73	min −1, max 2
**Everest (February)**	−36 ± 3 (7.0%)	40 ± 7 (18.1%)	−66 ± 3 (4.6%)	−1 ± 2 (126.2%)
	min −40, max −29	min 26, max 55	min −72, max −58	min −5, max 1
**K2 (February)**	−45 ± 2 (3.7%)	27 ± 4 (13.1%)	−75 ± 3 (3.5%)	1 ± 0 (49.0%)
	min −48, max −40	min 17, max 33	min −81, max −69	min 0, max 2

Temp, temperature; Wind, wind speed; WCT, wind chill equivalent temperature; FFT, facial frostbite time.

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
