# Peer review of "Comparison of Environmental Conditions on Summits of Mount Everest and K2 in Climbing and Midwinter Seasons"

_ijerph, 2021, doi:10.3390/ijerph18063040_

Round 1

Reviewer 1 Report

(1) The authors did not address the current relevant aspects of policies and existing laws/regulations of local authorities/governments which impact and regulate hiking in the two destinations.

(2) Mass of climber (line 122): the authors did not provide the rationale for use of 100 kg mass of climber. Is this the average mass of climbers over a period? The authors did not address gender or race of climbers. Are these characteristics of any relevance to the analysis?

(3) Climate Change - the scientific evidence suggests that global climate is being impacted by the climate change phenomenon. How does the potential impact of climate change influence or likely to influence the parameters/variables examined by the authors? The authors may wish to address the potential influence of climate change on their analyses and projections. if any.

(4) Citations (line 370) - should citations be corrected to Moore et al?

(5) Line 260: it may be useful to clarify the format of the value reflected (i.e. is the hyphen required or is a negative value (-326 hPa) intended?  This in not clear to me.

(6) Wind - 'Wind Speed' is discussed but no reference was made to 'Wind Direction'. Should this be of consideration?

(7) Precipitation - does precipitation levels and/or atmospheric moisture content have any relevance to the analysis? 

Author Response

Response to Reviewer 1 Comments

Dear Reviewer 1, 

Thank you for your valid and valuable comments on our article. We detail our responses to the questions and suggestions in the text below.

Point 1: The authors did not address the current relevant aspects of policies and existing laws/regulations of local authorities/governments which impact and regulate hiking in the two destinations.

Response 1: Our study aimed to assess and compare the weather at the summits of K2 and Everest in the climbing and the midwinter seasons. We did not address the hiking regulations in the two destinations because they are generally similar in Pakistan, Nepal and China and they do not significantly influence the number of climbers attempting both peaks. In each of the three countries the climber must have a permit to climb the mountain. The price of the permit depends on the mountain and the climbing season. Due to the Coronavirus pandemic, the issuing of climbing permits may be limited periodically.

Point 2: Mass of climber (line 122): the authors did not provide the rationale for use of 100 kg mass of climber. Is this the average mass of climbers over a period? The authors did not address gender or race of climbers. Are these characteristics of any relevance to the analysis?

Response 2: We used the formulas for VO2max and Speed presented by Matthews et al. [23] in which the mass of climber with equipment was set at 100kg. Matthews et al. followed the assumption made by West et al. [30] that the mass of a hypothetical climber of Everest equals 100 kg including equipment. West et al. [30] based their assumption on the average body weight of the climbers they analysed in their study and the mass of a standard climbing outfit and mountaineering equipment. Since the formulas for VO2max and Speed were based on research among Caucasian males, and because gender and race might affect VO2max and Speed changes at extreme altitudes, we have added (revised article,  line 124) that the results of VO2max and Speed represent average Caucasian male climbers. We have also added a reference to West et al’s work [30] in the revised article at line 126.

Point 3: Climate Change - the scientific evidence suggests that global climate is being impacted by the climate change phenomenon. How does the potential impact of climate change influence or likely to influence the parameters/variables examined by the authors? The authors may wish to address the potential influence of climate change on their analyses and projections. if any.

Response 3: We have addressed Reviewer 1’s concern about the effect of global warming on the weather parameters we examined by adding an explanatory paragraph to the Discussion (revised article, lines 290-296).

Point 4: Citations (line 370) - should citations be corrected to Moore et al?

Response 4: Thank you for spotting this mistake. We have corrected the citation to Moore et al.in the revised article (line 402).

Point 5: Line 260: it may be useful to clarify the format of the value reflected (i.e. is the hyphen required or is a negative value (-326 hPa) intended?  This in not clear to me.

Response 5: The hyphen was a typing error and was removed (line 271).

Point 6: Wind - 'Wind Speed' is discussed but no reference was made to 'Wind Direction'. Should this be of consideration?

Response 6: We estimated Wind Speed at the summits of Everest and K2,where there is no natural wind protection, e.g. by a rock formation or a slope Wind direction therefore does not affect the effect of Wind Speed on the climber. Wind direction and mountain topography should be considered when the wind is analysed on the different routes on a mountain. A route on the windward side of a mountain will expose a climber to more extreme wind than on a route on the leeward side. The analysis of weather on the various climbing routes on Everest and K2 was not the subject of our work.

Point 7: Precipitation - does precipitation levels and/or atmospheric moisture content have any relevance to the analysis? 

Response 7: We did not analyse precipitation levels and atmospheric moisture content because we focused on the climatic parameters that most limit human performance and survival. We agree that precipitation differences between seasons and mountains might will likely affect a climber’s speed: high precipitation levels decrease climbing speed due to difficult, deep snow conditions. Precipitation levels and atmospheric moisture content was beyond the scope of our work, but they are worth considering in future research focusing on the factors that influence climbing speed at extreme altitudes.

Reviewer 2 Report

Summary:

The purpose of this study was to compare different environmental parameters in climbing and midwinter season on K2 and Everest. Most environmental parameter were worse on Everest compared to K2, independent of the season. However, midwinter temperature was found to be more extreme on K2 than on Everest. Whereas winter conditions on K2 were more severe than Everest´s climbing season condition.

General  comments:

The manuscript requires editorial revision and language edition to enhance clarity and to eliminate grammatical and punctuation errors.

Non-scientific presentation of results, as data are not presented with standard deviation.

Major comments:

Abstract:

Please specify climbing and midwinter season for both summits (which month of the year were considered for the analyzation)

It is unusual to only report means without standard deviation. I recommend to present all data as mean ± SD.

Introduction:

Line 55: reference No 11 is not available

Line 55: “One climber, Nirmal Purja, reached the summit without supplemental oxygen” is this statement validated? If not, I would suggest to rewrite as follows: “One climber, Nirmal Purja has reported that he reached the summit without supplemental oxygen.”

Methods:

Line 85: please specify “in all month”

Line 86: Reference for climbing season on Everest and K2?

Results:

Please always provide SDs to every mean value that is presented.

The authors present the results in written form and the same results in Tables, which is redundant. I would suggest to present differences (% or absolute) in the text and Means ± SD in the tables.

Line 189: please check p= 1; that is not possible!

Figure 1: please indicate any significant difference between BP of the 2 summits.

Table 2: Please provide a legend with an explanation of abbreviations

Table 3: Table 3 is redundant, as the results are already presented in the text. Please consider to reorganize or to delete the table.

Discussion:

I would suggest to start with summarizing the main findings/the main goal of the study and further compare the results with previous research (start with line 226).

Same as for the result section: please also provide SDs when you present mean values.

Line 331: I would suggest to always start discussing the main question/goal of your study, the difference between K2 and Everest and than to start comparing your results with previous research.

Line 408: I suggest to delete “In accordance to our goals”, as it one would expect, that you conclude the main goal/findings of the study.

Author Response

Response to Reviewer 2 Comments

Dear Reviewer 2,

Thank you for your valid and valuable comments on our article. We detail our response to your questions and suggestions in the text below.

General  comments:

The manuscript requires editorial revision and language edition to enhance clarity and to eliminate grammatical and punctuation errors.

Response 1: The article has been re-edited and corrected for greater clarity in the text and to remove grammatical and punctuation errors.

Non-scientific presentation of results, as data are not presented with standard deviation.

Response 2: We have improved the presentation of our results by adding the standard deviation throughout the article. Detailed changes are described below.

Major comments:

Abstract:

Please specify climbing and midwinter season for both summits (which month of the year were considered for the analyzation)

Response 3: We now specify in the abstract which months we analyse and explain our choices (revised article, lines 20-21).

It is unusual to only report means without standard deviation. I recommend to present all data as mean ± SD.

Response 4: Standard deviations have been added to the mean values in the abstract (revised article, lines 26-32).

Introduction:

Line 55: reference No 11 is not available

Response 5: We have re-checked the availability of this article online by entering the website address listed in the References (reference No 11) (https://www.reuters.com/article/us-pakistan-mountain-k2/sherpas-successfully-complete-first-winter-summit-of-k2-spanish-climber-killed -idUSKBN29L0M2) and found that the article is freely available. Would Reviewer 2 kindly check the availability of the cited article again (revised article, line 57).

Line 55: “One climber, Nirmal Purja, reached the summit without supplemental oxygen” is this statement validated? If not, I would suggest to rewrite as follows: “One climber, Nirmal Purja has reported that he reached the summit without supplemental oxygen.”

Response 6: No mountaineering organization or committee validates ascents by confirming if the climb was accomplished with or without use of oxygen. Validation of the ascent is made by examining statements from the climber and witnesses and by analysing photographic or film documentation. Nirmal Purja’s first winter ascent of K2 without supplemental oxygen has not been challenged by any authority in the high-altitude community, so we would prefer to leave the sentence in its original form (revised article, lines 57-58).

Methods:

Line 85: please specify “in all month”

Response 7: We have improved the sentence and have specified that “all months” means in all months of the year from 1979 to 2019 (revised article, line 89).

Line 86: Reference for climbing season on Everest and K2?

Response 8: We have added references for the climbing seasons on Everest and K2 (revised article, line 90-93).

Results:

Please always provide SDs to every mean value that is presented.

Response 9: Standard deviations were added to all mean values in the results section (revised article, lines 145-213).

The authors present the results in written form and the same results in Tables, which is redundant. I would suggest to present differences (% or absolute) in the text and Means ± SD in the tables.

Response 10: We have followed your suggestion and have changed the Results section accordingly. We now present absolute differences in the text and Means ± SD in the tables (revised article lines 145-213).

Line 189: please check p= 1; that is not possible!

Response 11: The result was checked and found to be p=0,99996. This finding was changed in the text (revised article, line 197).

Figure 1: please indicate any significant difference between BP of the 2 summits.

Response 12: BP values for the same months on the two summits differ significantly (p <0.001). We have added an explanatory sentence to the caption of Figure 1 (revised article, lines 176-177).

Table 2: Please provide a legend with an explanation of abbreviations

Response 13: The have added a legend as suggested (revised article, line 188). We have also added a legend to Table 1 (revised article, line 171).

Table 3: Table 3 is redundant, as the results are already presented in the text. Please consider to reorganize or to delete the table.

Response 14: We decided to delete the table (revised article, lines 223-226).

Discussion:

I would suggest to start with summarizing the main findings/the main goal of the study and further compare the results with previous research (start with line 226).

Response 15: We have reorganized the Discussion according to Reviewer 2’s suggestions (revised article, lines 229-240).

Same as for the result section: please also provide SDs when you present mean values.

Response 16: We have added standard deviations to the mean values in the Discussion as suggested (revised article, lines 229-308).

Line 331: I would suggest to always start discussing the main question/goal of your study, the difference between K2 and Everest and than to start comparing your results with previous research.

Response 17: We have reorganized the Discussion as suggested by Reviewer 2 (revised article, lines 351-363).

Line 408: I suggest to delete “In accordance to our goals”, as it one would expect, that you conclude the main goal/findings of the study.

Response 18: We have deleted this fragment as suggested (revised article, line 441).

Round 2

Reviewer 2 Report

I thank the authors for their rebuttal. I have just one litte thing to check:

line 278-279: "The prediction of the future BP levels on the summits of Everest and K2 should consider they will likely increase due to global climate warming." please check this sentence...

Thank you very much!

Author Response

Response to Reviewer 2 Comments, Round 2

Dear Reviewer 2,

Thank you for your positive opinion concerning our rebuttal and your previous valid and valuable comments on our article.

Comment:

line 278-279: "The prediction of the future BP levels on the summits of Everest and K2 should consider they will likely increase due to global climate warming." please check this sentence...

Response:

In our opinion above sentence does not need to be changed. It has been observed by other authors that global warming increases the annual mean BP near Everest’s summit [23,43].  We suggest that, the prediction of future BP levels on the summits of Everest and K2 should take this fact  into account.

Thank You very much.